# Targeting NF-κB Signaling for Multiple Myeloma

**DOI:** 10.3390/cancers12082203

**Published:** 2020-08-06

**Authors:** Ada Hang-Heng Wong, Eun Myoung Shin, Vinay Tergaonkar, Wee-Joo Chng

**Affiliations:** 1Laboratory of NF-κB Signaling, Institute of Molecular and Cell Biology (IMCB), Agency for Science, Technology and Research (A*STAR), Singapore 138673, Singapore; shinem@imcb.a-star.edu.sg (E.M.S.); vinayt@imcb.a-star.edu.sg (V.T.); 2AW Medical Company Limited, Macau, China; 3Department of Pathology, Yong Loo Lin School of Medicine, National University of Singapore, Singapore 119074, Singapore; 4Department of Centre for Cancer Biology, University of South Australia and SA Pathology, Adelaide, SA 5000, Australia; 5Cancer Science Institute of Singapore, Singapore 117599, Singapore; 6Department of Medicine, Yong Loo Lin School of Medicine, National University of Singapore, Singapore 119228, Singapore; 7Department of Hematology-Oncology, National University Cancer Institute of Singapore, National University Health System, Singapore 119074, Singapore

**Keywords:** multiple myeloma, NF-κB, biomarker-guided targeted therapy, precision medicine

## Abstract

Multiple myeloma (MM) is the second most common hematologic malignancy in the world. Even though survival rates have significantly risen over the past years, MM remains incurable, and is also far from reaching the point of being managed as a chronic disease. This paper reviews the evolution of MM therapies, focusing on anti-MM drugs that target the molecular mechanisms of nuclear factor kappa B (NF-κB) signaling. We also provide our perspectives on contemporary research findings and insights for future drug development.

## 1. Background

Multiple myeloma (MM) is the second most common hematologic malignancy, accounting for 1% of all cancers globally in 2016 [1]. MM mainly occurs in older people, with a median age of 66–70 years at the time of diagnosis [1]. Age demography significantly contributes to MM progression and regimen design [2,3]. Survival rates are higher in younger people, partially due to the feasibility of autologous stem cell transplant (ASCT) and better drug tolerability during adjuvant therapy to ASCT or non-transplant systemic therapy. Nevertheless, overall survival rates have risen significantly over the past decades [4,5], with a five-year survival rate of 53.9% between 2010 and 2016 in the United States (US) [5]. Improved outcomes resulted from a significant increase of the use of novel therapies, including proteasome inhibitors and immunomodulatory drugs (IMiDs), which increased from 8.7% in 2000 to 61.3% in 2014 in US patients [6]. Additionally, the introduction of low-dose continuous therapy and maintenance therapy schemes post-ASCT or high-dose induction therapy also contributed to improved clinical outcomes, mainly through palliating adverse effects to increase drug tolerability, especially in weak elderly patients [7]. Aside from improvements in therapeutic design, novel drugs are designed to target specific molecular mechanisms involved in MM, especially the nuclear factor kappa B (NF-κB) signaling pathway [8,9,10], which is described in the following context.

## 2. Multiple Myeloma and NF-κB Signaling

Myeloma, also known as plasma cell myeloma, is the accumulation of malignant plasma cells in the bone marrow. Initially, in the asymptomatic phase (known as smoldering myeloma), myeloma cells produce abnormally substantial amounts of monoclonal proteins (M-proteins) that are released to the blood stream. As the disease progresses to the symptomatic phase, known as MM, the myeloma cells harness the bone marrow microenvironment to promote growth and invasion. MM progression leads to bone destruction, hypercalcemia and renal insufficiency, and may result in patient lethality.

MM is linked to the frequent onset of hyperploidy, chromosomal aberrations, genetic mutations and epigenetic transformations. Among these anomalies, hyperploidy is most prevalent, contributing to nearly half of all MM cases [11]. In non-hyperploid MM, chromosomal aberrations at the B cell class switching gene locus *IGH* frequently occur, leading to aberrant production of M-proteins in MM patients [12,13]. Aside from chromosomal changes, genetic mutations common in cancers frequently occur in MM too, e.g., the oncogenic *KRAS* and *NRAS* transformations and loss-of-function *TP53* mutations [11,14,15,16]. Some genetic modifications are more MM-specific and contribute to hyperactive NF-κB signaling. These include the amplification or rearrangement of *NIK*, *LTBR*, *TACI*, *NFKB1*, *NFKB2* and *CD40* genes, as well as deletion or loss-of-function mutations in genes like *CYLD*, *BIRC2/BIRC3* (*cIAP1/cIAP2*), *TRAF2* and *TRAF3* [17]. On the other hand, MM progression displays distinct epigenetic landscape changes. For example, extensive DNA hypomethylation in non-CpG islands occurs during the transition from monoclonal gammopathy of undetermined significance to the myeloma stage [18]. Moreover, hypermethylation of a subset of transcription factors, e.g., *FOXD2*, *GATA4*, *RUNX2*, and cell cycle-related genes, e.g., *CDKN2B*, potentially remodels cellular processes to promote tumorigenesis [18]. Furthermore, MM development is supported by the promoter methylation of the *P53* gene, which is sustained by the NF-κB-regulated cytokine interleukin-6 (IL-6) [19]. Aside from DNA methylation, histone modifications such as acetylation and methylation also significantly alter the epigenetic landscape and drug response of MM [20]. For instance, overexpression of the histone methyltransferase gene *EZH2* that frequently occurs in MM may be induced by hyperactive non-canonical NF-κB signaling [21]. Inhibition of EZH2 sensitizes MM to bortezomib treatment in vivo, through cooperative *MYC* suppression and inhibition of H3K27 trimethylation to regulate genes involved in B cell metabolism and antibody production [22,23]. NF-κB gene mutations are known to be the most prevalent in MM among all human cancers [14,15,17,24,25], and plays a pivotal role in anti-cancer therapy and drug resistance [26,27,28,29].

NF-κB refers to a family of transcription factors that form homo- and hetero-dimers within the family, as well as with other transcription factors [30]. NF-κB signaling is classified into the canonical and non-canonical pathways that are represented by the transcriptional protein complexes of p50/RelA and p52/RelB, respectively [31]. These two pathways are activated by distinct membrane receptors that respond to extracellular ligands like tumor necrosis factor α (TNFα), interleukin-1 (IL-1), receptor activator of NF-κB ligand (RANKL), and so on (Figure 1). In canonical NF-κB signaling, receptor activation leads to formation of the TRAF2-TRAF5-TRAF6 complex, which activates TAK1 kinase to phosphorylate the complex of IKKα, IKKβ and NEMO. IKK complex phosphorylation subsequently triggers the degradation of IκB to release the p105 protein for proteasomal processing to the p50 protein. Consequently, the p50/RelA complex translocates to the nucleus and initiates transcription. Non-canonical NF-κB signaling involves the TRAF2-TRAF3-TRAF6 complex, which activates the NIK kinase to phosphorylate the IKKα kinase. Phosphorylated IKKα then triggers proteasomal processing of p100 to p52 for transcriptional activation. Although the canonical and non-canonical pathways have variant triggering signals and downstream targets, both pathways are involved in MM pathogenesis and progression [15,24,25].

NF-κB signaling plays a pivotal role in promoting cancer growth, angiogenesis and tumor-microenvironment crosstalk, which mainly involves the production of pro-inflammatory cytokines, inflammation mediators, cell adhesion molecules, among others, to establish a favorable tumor microenvironment for MM tumorigenesis and disease progression. Non-canonical NF-κB signaling is also a key determinant of other oncogenic drivers, such as telomerase and telomeric proteins, which are commonly deregulated in cancers [32,33,34,35]. NF-κB signaling, in combination with other potent transcription factors such as STAT3, also plays important roles in regulating apoptosis and polarization of immune subtypes, which contribute to a pro-tumoral microenvironment [36,37]. Hence, many first-line anti-MM drugs have an indirect impact on the NF-κB signaling pathway (Figure 1). For instance, bortezomib is a reversible inhibitor of the 26S proteasome [38] and thus prevents the proteasomal cleavage of NF-κB proteins and the IκB protein to inhibit gene transcription activation. The insult of bortezomib on MM cells is further enhanced by the fact that the proteasome is overloaded by excessive M-protein production in myeloma cells. On the other hand, the corticosteroid dexamethasone induces inhibitor of κB (IκB) protein synthesis to inhibit NF-κB signaling [39]. Another first-line therapy drug, the IMiD lenalidomide, diminishes sustained RelA binding to open chromatin by inhibiting the Ikaros proteins Ikzf1 and Ikzf3 [40,41,42]. Co-administration of lenalidomide and dexamethasone suppresses interleukin-2 (IL-2), immunoglobin M (IgM) and immunoglobin G (IgG) production, hence reducing the protein load of MM patients [43]. In recent years, immunotherapy using antibodies and chimeric antigen receptor T (CAR-T) cells against specific NF-κB signaling receptors has gained more attention. For example, antibodies against B cell-activating factor (BAFF) inactivate non-canonical NF-κB signaling in MM cells [44,45]. Clinical trials are ongoing, so it is still too early to conclude whether any of these strategies works.

In addition to intracellular NF-κB hyperactivation, MM cells also manipulate NF-κB signaling in the bone marrow microenvironment to promote cancer growth and invasion (Figure 2). On the one hand, myeloma cells hijack stromal cells to secrete cytokines, such as interleukin-6 (IL-6), receptor activator of NF-κB ligand (RANKL) and vascular endothelial growth factor (VEGF), to promote cancer proliferation and angiogenesis [46]. On the other hand, myeloma cells secrete Dickkopf-1 (Dkk1) and macrophage inflammatory factor 1α (MIP-1α) to inhibit osteoblast differentiation to block new bone formation, and activate osteoclasts to promote osteolysis [47,48]. In late-stage MM patients, the myeloma cells acquire additional genetic abnormalities that lead to reduced dependency on the microenvironment (e.g., *P53* mutation), increased drug resistance and increased aggressiveness of the clone (e.g., 1q21 amplification and *CKS1B* overexpression) [49]. Hence, therapies against both MM cells and microenvironment control, such as daratumab, an antibody drug against CD38 that induces antibody-dependent cytotoxic events in CD38-expressing cancer cells and complement-dependent cytotoxicity [50,51,52], show success in MM treatment [53].

Even though receptor-/ligand-specific antibodies can target NF-κB signaling with high specificity, the diverse NF-κB signals in MM cells and their microenvironment limit the application of these therapies to treat MM effectively in vivo. Tumor evolution also contributes to altering pathways to develop drug resistance. Hence, directly targeting the molecular machinery of NF-κB signaling remains crucial. Until now, no specific NF-κB inhibitor has been approved for treating MM. We will discuss the challenges of developing specific NF-κB inhibitors for MM treatment.

## 3. NF-κB Signaling: The Rose with Thorns in MM Treatment

Even though NF-κB signaling plays a critical role in MM, specifically targeting this signaling pathway proves to be more difficult than previously thought.

The foremost hurdle is drug safety. NF-κB signaling plays a key role in innate immunity and inflammation. Constitutive inactivation of NF-κB signaling silences the immune system, subsequently rendering patients susceptible to infections. Population-based studies have pointed out that MM patients displayed a seven-fold higher risk of bacterial infection and a 10-fold higher risk of viral infections as compared to randomized control individuals, resulting in a stunning 22% death rate among MM patients at one-year follow-up [54]. It is noteworthy that transplanted patients displayed a broader spectrum of infection [55], where infection rate may be reduced by combined IMiD therapy [56], due to the fact that immunosuppressive drugs are administered to prevent graft-versus-host defense. On the other hand, bortezomib-based therapy is associated with a higher risk of severe infection in various studies [56,57,58], possibly through inhibition of both canonical and non-canonical NF-κB pathways. Hence, it is hypothesized that targeting of one NF-κB pathway may be safer than inactivating both NF-κB signaling pathways. Nevertheless, trials of many IKKβ inhibitors showed severe adverse effects [59], even though non-canonical NF-κB signaling is hypothesized to be unaffected. In contrast, the anti-RANK antibody denosumab, which is recommended as adjuvant therapy to MM patients to alleviate hypercalcemia due to hyperactive osteoclasts [60], produces little toxicity but cannot treat MM because of its narrow-spectrum inhibition of RANK-mediated non-canonical NF-κB signaling. Hence, striking the right balance between drug safety and treatment efficiency remains challenging.

Secondly, the context-specific and spatio-temporal regulation of NF-κB signaling complicates therapeutic design. This complexity is further complicated by the interplay between cancer and its immune microenvironment [26,46,61,62]. For example, the non-steroidal anti-inflammatory drugs (NSAIDs) (e.g., aspirin, sulindac and tolfenamic acid) that target the *COX-1/2* genes may suppress NF-κB signaling during short-term administration but activate NF-κB signaling after prolonged treatment [59].

Thirdly, selectivity remains a critical issue. For instance, several IKKβ inhibitors exhibit off-target effects, whereas others display high selectivity toward IKKβ and loss of inhibition through IKKα [59]. In this regard, combined treatment may offer hope for highly selective inhibitors, but this also requires more consideration of the treatment burden on patients, especially when many MM patients are old and weak. Nevertheless, high selectivity suffers from poor treatment efficiency. For instance, the anti-BAFF antibody tabalumab, which specifically inhibits non-canonical NF-κB signaling, failed to improve outcomes in MM patients in a phase II trial in which it was combined with bortezomib and dexamethasone [63].

Lastly, the lack of structural information hampers rational drug design, although solving the structure does not guarantee success either. For example, structure of the NF-κB inducing kinase (NIK) has been solved [64], but the NIK inhibitors AM-0216 and AM-0561 failed due to poor pharmacokinetics in vivo, even though they displayed high NIK binding affinity and *NIK*-dependent cytotoxicity in vitro [65]. On the contrary, a peptide mimetic of the NF-κB essential modulator (NEMO) binding domain that blocks IKK complex formation [66] made its way to clinical trials in dogs for treating diffuse large B cell lymphoma (DLBCL) and soft tissue sarcoma (STS) [67], but no clinical trial has been reported to treat humans. Other molecular targets might lack specific inhibitors after solving the structure [68]. Consequently, no specific NF-κB inhibitor has been successfully developed to treat MM until now.

## 4. MM Therapy: A Steep Road to Success

MM therapy mainly involves stem cell transplant and systemic drug therapy with occasional use of radiotherapy for plasmacytomas (Figure 3). ASCT is an important treatment modality for MM patients, whereas allogeneic stem cell transplant is suggested for high-risk patients with complex karyotypes [69]. However, allogeneic stem cell transplant is still rarely used in MM patients because of high rates of treatment-related mortality. Furthermore, stem cell transplant is often infeasible because of the frailty of MM patients, who are mostly elderly, aged 65 and above. Hence, systemic drug therapy remains an indispensable approach for MM treatment. Similar to many other cancers, MM is often treated by a combination of drugs.

Among these drugs, bortezomib is the most frequently used drug for MM treatment. Bortezomib is commonly paired with IMiDs such as lenalidomide and thalidomide, or alkylating agents such as melphalan, cyclophosphamide and dexamethasone. Dexamethasone, lenalidomide and prednisone all exhibit anti-inflammatory responses. On the other hand, melphalan is a DNA alkylating drug that has been used to treat MM since the 1950s [70]. Among all these drugs, melphalan and prednisone are the oldest drugs used in MM treatment [71]. It is noteworthy that combination therapy is usually used for MM treatment, often as a means of exploiting drugs with different mechanisms of action.

Existing anti-MM drugs target a diverse array of molecular pathways (Table 1). Analysis of the recommended drugs along the lines of therapy, together with those undergoing clinical trials, displays an interesting trend (Figure 4). Briefly, anti-inflammatory drugs take a central role in MM therapy and are applied to both standard therapy and maintenance therapy for MM patients, given the pivotal role of NF-κB and inflammation in MM progression. Next, the impact of protein secretion load in myeloma cells and the resultant endoplasmic reticulum stress make the cells vulnerable to inhibition of the proteolysis pathways. Therefore, proteasome inhibitors, including bortezomib and carfilzomib, often make up a base for different combination therapies. Aside from the MM-specific mechanisms, drugs that disrupt DNA and RNA synthesis, promote cell death and inhibit angiogenesis are frequently applied to MM treatment too. However, the most intriguing part is the increasing trend of other therapeutic mechanisms in drugs undergoing clinical trials or in research. The common mechanism of these drugs is the specific targeting of certain molecules or molecular pathways, especially NF-κB signaling. For instance, selinexor blocks exportin to retain NF-κB in the nucleus [72,73].

Targeting NF-κB signaling takes the center stage of anti-MM drug development. For instance, antibodies against CD38, CS1, BAFF and BCMA showed success in MM treatment [50,53,74,75]. Parallel tests were conducted using chimeric antigen receptor T (CAR-T) cells [76,77,78], but relapse became a major concern [79]. In addition, antibodies and peptides against the programmed cell death 1 (PD-1) protein and its ligand PD-L1 are also heavily investigated [80], where PD-L1 can be induced by canonical NF-κB signaling [81]. Development of IKK inhibitors persists [59], whereas development of NIK inhibitors is also ongoing [82,83]. Recently, a novel NIK inhibitor called Cpd33 is reported to inhibit RANKL-induced osteoclastogenesis in an ovariectomized mouse model [84]. Oral administration of another NIK inhibitor XT2 in mice relieved toxin-induced liver inflammation [85]. These inhibitors may be tested in MM mouse models in the future. Progress is also made to disrupt NF-κB-mediated crosstalk between the tumor and its microenvironment. For example, an antibody against Dkk1 is proposed to alleviate osteolysis [86].

## 5. Future Direction: Biomarker-Guided Targeted Therapy

The first paradigm shift in MM treatment occurred in early 2000s with the introduction of the first-in-class proteasome inhibitor bortezomib [71]. Research on the correlation between proteasome load and degradative capacity on the sensitivity of MM cells toward bortezomib was conducted [87], suggesting that the levels of M-proteins and proteasome expression can be biomarkers for proteasome inhibitor sensitivity. This incidence marks the start of evidence-based therapeutic design for MM treatment. However, even though bortezomib is highly effective, its severe neurotoxicity is often intolerable, leading to drop-offs in many studies [88]. This subsequently led to exploration of the oral alternative ixazomib, which exhibits lower neurotoxicity [89]. Additionally, second generation proteasome inhibitors with different chemical moieties were developed to improve outcomes and reduce adverse reactions. For instance, carfilzomib showed a better response than bortezomib in refractory MM [90,91,92]; its major adverse effects on the cardiac system also occurred in fewer patients [93]. These protocols have paved the way for more precise regimens for application to different patient subgroups.

Recently, Shin, et al. identified lymphocyte cytosolic protein 1 (*LCP1*) gene as a novel NF-κB target in *TRAF3* or *NIK* mutant MM cells [94] (Figure 5). *TRAF3* loss induces constitutively active NIK [95,96], which frequently occurs in MM patients [8,14,17]. B cell-specific *TRAF3*^-/-^ mice developed MM and peripheral B cell hyperplasia [97,98,99,100].

*LCP1* encodes for the L-plastin protein, which is required for sealing ring formation in osteoclasts [101], but not in bone formation by osteoblasts in vitro [102]. Secreted L-plastin by breast cancer cells mediates metastatic osteolysis in mice [103]. Despite its osteolytic function, L-plastin contributes to metastasis of breast cancer, melanoma and colon cancer [103,104,105,106,107,108,109]. Studies also reported that L-plastin is responsible for disease progression of bladder and kidney cancer [110,111], and homing of chronic lymphocytic leukemia (CLL) to bone marrow [112]. Analysis of transcriptomic data showed that *LCP1* overexpression is significantly correlated with poor overall and progression-free survival in MM patients [94].

L-plastin not only serves as a prognostic biomarker, but may also serve as a therapeutic target. For instance, radiotherapy represses exosomal release of L-plastin in both the tumor and its niche, producing radiation-induced bystander effects and enhance outcomes [113]. Inhibition of L-plastin Ser5 phosphorylation re-sensitizes resistant MM cells to IMiDs and proteasome inhibitors [114]. Collectively, these data suggest that inhibiting L-plastin provides an alternative route to treating MM, specifically in *TRAF3*-mutant MM patients. In this regard, peptide inhibitors against L-plastin that inhibit osteoclast activity [102,115] might be tested. Alternatively, NIK inhibitors under development [82,83], or shown to repress inflammation in other mouse models [84,85], may be tested in MM mouse models too.

The revelation of *LCP1′*s involvement in *TRAF3*-mutant MM has led to deeper investigation of the role of calcium signaling in MM development and progression. It has been reported that calcium signaling activates NF-κB in B cells [116]. For instance, B cell receptor (BCR) activation stimulates store-operated calcium entry (SOCE) to induce *NFKB2* expression [117], which encodes for p100 that is processed into p52 during non-canonical NF-κB activation. On the other hand, calcineurin A associates with TRAF3 and NIK to inhibit LTβR-mediated NIK activity and TWEAK-mediated p100 processing in fibroblasts [118]. Simultaneously, the calcineurin A-specific inhibitors cyclosporin A and FK506 inhibit IgM-induced B cell activation and trigger cell death [119]. Taken together, targeting L-plastin and calcium-dependent NF-κB activation may be favorable for MM therapy.

Massive chromosomal aberrations are a key feature of MM. Nearly half of MM tumors are hyperdiploid, usually comprising multiple copies of chromosomes 3, 5, 7, 9, 11, 15, 19 and 21 [11], whereas non-hyperdiploid MM tumors usually contain *IGH* translocations that do not involve c-myc [12,13]. This finding led to the proposal of patient risk group stratification based on molecular signatures of MM [11]. However, cytogenetic analysis of all known mutations might be unaffordable for the less-privileged, so resource-stratified guidelines are proposed to circumvent the cost issue [120]. In order to carry out patient risk group stratification, genetic test was suggested to be mandated as a routine clinical test for MM patients and subsequent clinical trials in 2009 [121]. Cytogenetic analysis is exploited to indicate allelic mutations relevant to regimen design. For example, homozygous carriers of the *NFKB1* -94insATTG polymorphism were retrospectively demonstrated to benefit more from bortezomib treatment than MM patients carrying the deletion allele [122]. Bortezomib also exhibits a good response in del17p but not t (4;14) patients and those with 1q21 gain, although its combination with lenalidomide and dexamethasone is promising in del17p, del1p, t (4;14), and t(14;16) patients [7]. Venetoclax is highly effective in t (11;14) MM patients [123]. Furthermore, epigenetic changes have been correlated to certain cytogenetic subgroups. For example, extensive DNA hypomethylation is correlated to MM subgroups of hyperploidy, t (4;14), t (11;14), and t (14;16) translocations, but not the del1p, gain 1q, del13q, del16q, del17p and del22q subgroups [18]. Of note, t (4;14) translocation upregulates the histone methyltransferase multiple myeloma SET domain protein (MMSET) [124] and inhibition of MMSET activity inhibits MM cell proliferation in vitro [125]. Nevertheless, more research is needed to elucidate the correlation between different karyotypes and drug sensitivity.

Lastly, drugs to delay MM progression and prevent or re-sensitize MM to treatment are being investigated. For example, an antibody against IL-6 has been tried in smoldering myeloma patients to delay MM progression [126]. In addition, a potent multi-drug resistance modulator, valspodar, has been investigated to circumvent decreased drug deposition due to P-glycoprotein overexpression in response to therapy [127], but failed because of unimproved treatment outcomes and increased toxicity [128]. In contrast, nelfinavir sensitizes MM to overcome proteasome inhibitor resistance through modulating TCF11/Nrf1-mediated proteasome recovery [129]. Hence, drug development is striving incessantly to overcome the ever-evolving forms of drug resistance.

In addition to treatment efficacy, the tendency to trigger secondary neoplasm development is another key factor to consider during development of novel anti-MM drugs. Early small cohort studies showed that DNA alkylating drugs might possess a higher tendency to induce secondary neoplasms [130,131,132]. However, more recent study using a 403-patient cohort suggested that the therapeutic mechanism exhibited an insignificant impact; instead, complex karyotypes are largely correlated to the risk of developing secondary neoplasms [133]. Nevertheless, the skewed tendency of developing myelodysplastic syndrome (MDS) [130] warrants deeper investigation into therapy-related secondary neoplasms.

## 6. Conclusions

MM therapy has come a long way, evolving from the miscellaneous application of rhubarb and orange peel to the anti-proliferative melphalan and anti-inflammatory prednisone, to the first-in-class proteasome inhibitor bortezomib [71]. History has shown that MM treatment strategies evolved with our understanding of the molecular mechanism of MM progression. In this era of precision medicine, evidence-based therapeutic design has become the golden rule for regimen design. Regimen design considers factors like patient characteristics, therapeutic efficacy, working principle and adverse effects, among many others, from the scientific perspective. From the ethical point of view, treatment burden, quality of life and healthcare cost have received more attention in recent years. In parallel to treatment, diagnostic tests to identify biomarkers and predict treatment outcomes are routinely conducted in some countries. We predict that biomarker-guided targeted therapy will expand as our knowledge grows and continues to evolve in the next decade.

## Figures and Tables

**Figure 1 cancers-12-02203-f001:**
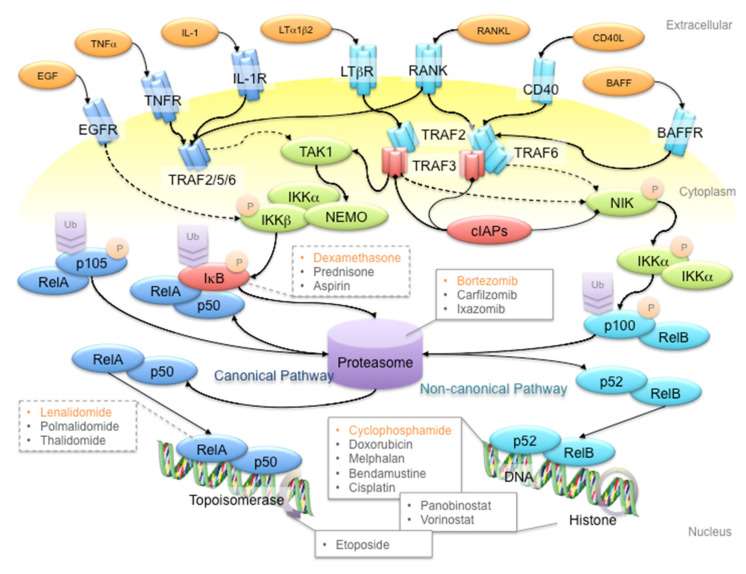
Schematic diagram of the NF-κB signaling pathway and anti-multiple myeloma (MM) drug targets. First-line anti-MM drugs (highlighted in orange) passively target the canonical and/or non-canonical pathways to shut down NF-κB signaling. For example, bortezomib inhibits the 26S proteasome to hinder the processing of p105 and p100 proteins, to prevent gene transcription activation in canonical and non-canonical NF-κB signaling pathways, respectively; dexamethasone induces IκB protein synthesis to inhibit p105 processing; lenalidomide reduces RelA binding to open chromatin; cyclophosphamide is a DNA alkylating agent that disrupts DNA replication and genome stability. Ligands, adaptor proteins and transcriptional complexes involved in canonical NF-κB signaling are depicted in dark blue, whereas those involved in non-canonical NF-κB signaling are depicted in light blue; kinases are depicted in green; inhibitors are depicted in red. P and Ub indicate the post-translational modifications of phosphorylation and ubiquitination, respectively. Arrows with triangle heads indicate activation, whereas arrows with rhomboid heads indicate inactivation/inhibition; direct interactions are indicated by solid lines, whereas indirect interactions are indicated by dash lines.

**Figure 2 cancers-12-02203-f002:**
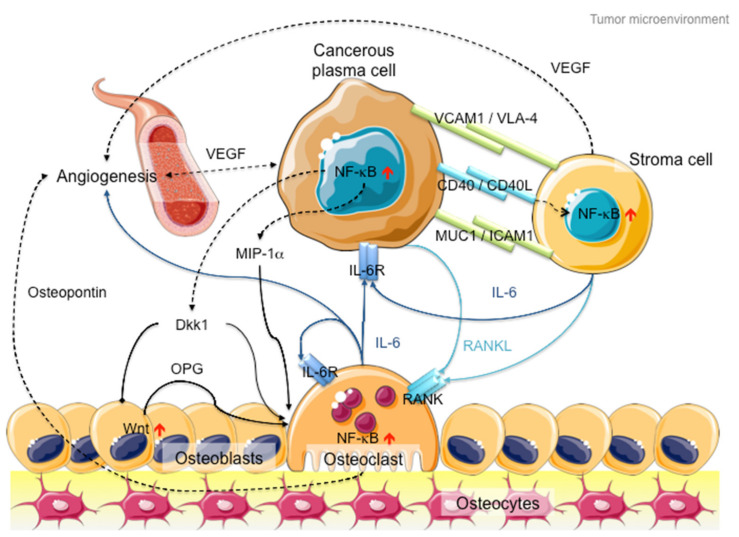
Schematic diagram of the MM microenvironment. Cancerous plasma cells interact with stroma, osteoblasts and osteoclasts through membrane receptor interactions and secretory cytokine pathways. Hyperactive NF-κB signaling plays a pivotal role in disease progression through transcriptional activation of the secretion of various cytokines like IL-6, RANKL and Dkk1 to promote cancer cell proliferation, osteoblast inactivation, osteoclast hyperactivation and angiogenesis. Proteins involved in canonical NF-κB signaling are indicated in dark blue, whereas proteins involved in non-canonical NF-κB signaling are indicated in light blue. Arrows with triangle heads indicate activation, whereas arrows with rhomboid heads indicate inactivation/inhibition; direct interactions are indicated by solid lines, whereas indirect interactions are indicated by dashed lines. Artistic images were downloaded from Servier Medical Art (https://smart.servier.com/; Servier Medical Art by servier is licensed under a creative commons attribution 3.0 unported license).

**Figure 3 cancers-12-02203-f003:**
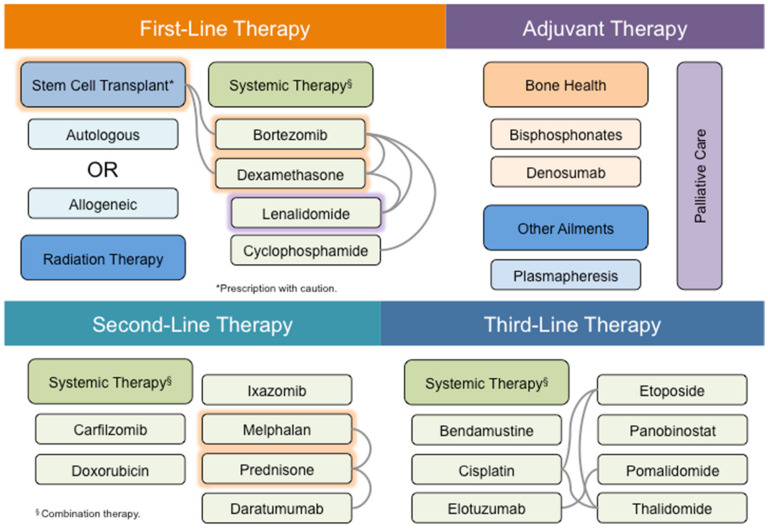
Therapeutic options of MM. Systemic therapy is the most common treatment option for MM patients. Systemic therapy is usually administered in a combined manner, with synthetic drugs and antibodies (green box), or in conjunction with stem cell transplant (cyan box). For example, bortezomib and dexamethasone constitute the adjuvant therapy for stem cell transplant patients in first-line therapy (shadowed in orange and connected by gray lines). On the other hand, melphalan and prednisone are recommended for non-transplant patients in first-line therapy (shadowed in orange and connected by gray lines). Alternatively, lenalidomide may be recommended with bortezomib and dexamethasone in non-transplant patients in first-line therapy (connected by gray lines) or administered alone during palliative care (shadowed in purple). Therapeutic options are color-coded in boxes according to their application and placed or shadowed under the categories of first-line (orange), second-line (cyan) and third-line (blue) therapy, or palliative care (purple), according to the frequency of recommendation for clinical use under the National Comprehensive Cancer Network (NCCN) guidelines. Combination therapies on the same line of therapy are connected by gray lines, whereas those spanning across different lines of therapy are not indicated.

**Figure 4 cancers-12-02203-f004:**
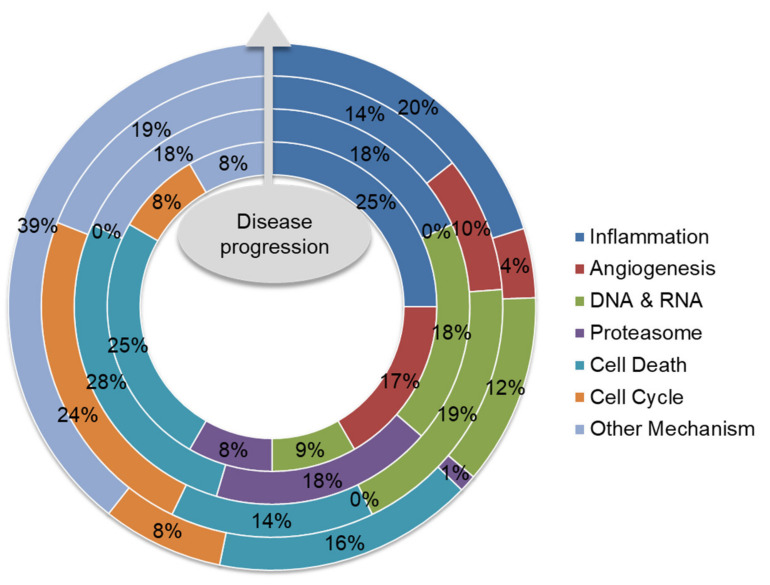
Summary of anti-MM drug mechanisms along the lines of therapy. Systemic therapy with synthetic drugs and antibodies constitutes the majority of anti-MM treatment options, and is often applied in a combined manner. In combination therapy, drugs that exhibit different mechanisms of action are applied. For example, bortezomib, which targets the proteasome, is administered with dexamethasone, which exhibits an anti-inflammatory response. Sequential administration of drugs from first-line (innermost circle) to third-line (3rd circle) therapy is prescribed as disease progresses; drugs under clinical trial (outermost circle) are also analyzed. Even though proteasome inhibitors occupy a small percentage along the lines of therapy, they are often used as a base in combination therapy. On the other hand, IMiDs and death-inducing drugs constitute a huge percentage along the lines of therapy. All drugs are classified into first-line (innermost circle), second-line (2nd circle) and third-line (3rd circle) therapies, according to the frequency of recommendation for clinical use under the National Comprehensive Cancer Network (NCCN) guidelines; drugs under active clinical trial as exemplified in Table 1 are also shown (outermost circle). Drug mechanisms are demonstrated in the graph legend; the percentages indicate the percentage of drugs displaying the corresponding mechanism, compared to all drugs in the specified line of treatment.

**Figure 5 cancers-12-02203-f005:**
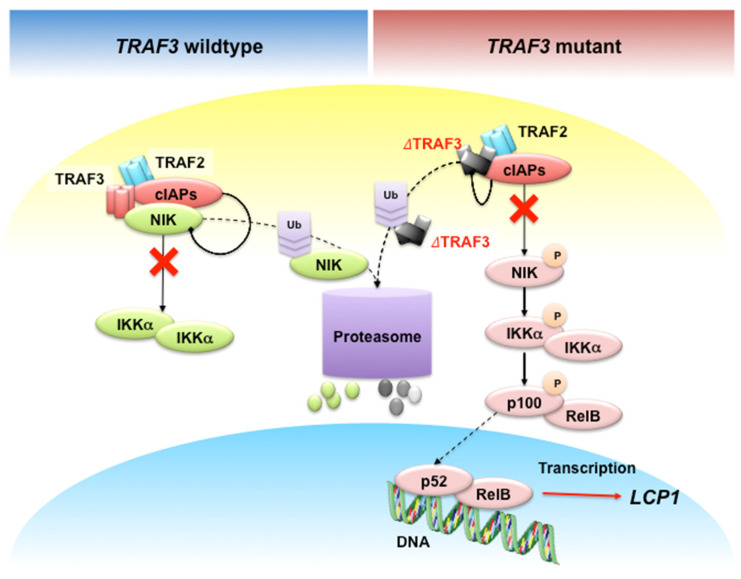
Constitutive NIK-driven NF-κB activation without functional TRAF3 in MM. In normal physiological conditions, TRAF3-TRAF2 interaction recruits the E3 ligases cIAPs to NIK to induce K48 ubiquitination, followed by proteasome-dependent degradation of NIK. In *TRAF3*-mutant MM cells, mutations in the TRAF or MATH domain of *TRAF3* result in ΔTRAF3 proteins that are unable to interact with TRAF2 and NIK, thus stabilizing NIK to activate non-canonical NF-κB signaling. Arrows with triangle heads indicate activation, whereas arrows with rhomboid heads indicate inactivation/inhibition; direct interactions are indicated by solid lines, whereas indirect interactions are indicated by dashed lines.

**Table 1 cancers-12-02203-t001:** Summary of anti-MM drug mechanisms.

Drug Name	Line of Therapy	Inflammation	Angiogenesis	DNA & RNA	Proteasome	Cell Death	Cell Cycle	Other Mechanism	Drug Target
Bortezomib	1								NF-κB
Dexamethasone	1								
Lenalidomide	1								VEGF, bFGF
Cyclophosphamide	1								DNA
Carfilzomib	2								Proteasome
Daratumumab	2								CD38
Doxorubicin	2								Topoisomerase
Ixazomib	2								Proteasome
Melphalan	2								DNA
Prednisone	2								
Bendamustine	3								DNA
Cisplatin	3								DNA
Elotuzumab	3								CS1
Etoposide	3								Topoisomerase
Panobinostat	3								HDAC
Pomalidomide	3								VEGF, bFGF
Thalidomide	3								VEGF, bFGF
Abatacept	CT								CD80, CD86
Abemaciclib	CT								CDK4, CDK6
Acalabrutinib	CT								BTK
ACP-319	CT								PI3K
ALT-803	CT								IL-15
ASA	CT								COX-1/2
Atezolizumab	CT								PD-L1
Avelumab	CT								PD-L1
Azacitadine	CT								DNA methylation
AZD5991	CT								Mcl-1
Binimetinib	CT								MEK-1/2
Busulfan	CT								DNA
Carmustin	CT								DNA
CC-92480	CT								CRBN
CCS1477	CT								p300, CBP
Cetrelimab	CT								PD-1
Clarithromycin	CT								Antibiotic
CLR131	CT								
Cobimetinib	CT								MEK1
CT-011	CT								PD-1
CYT-0851	CT								RAD51
Cytarabine	CT								DNA
Dabrafenib	CT								BRAF
Denosumab	CT								RANKL
Depsipeptide	CT								HDAC
Durvalumab	CT								PD-L1
Enasidenib	CT								IDH2
Encorafenib	CT								BRAF
Erdafitinib	CT								pan-FGFR
Fludarabin	CT								DNA
GBR1342	CT								CD38, CD3
Gemcitabine	CT								DNA
GSK2857916	CT								BCMA
GSK3174998	CT								OX40
GSK3359609	CT								ICOS
Idasanutlin	CT								MDM2
Ipilimumab	CT								CTLA4
Isatuximab	CT								CD38
JNJ-42756493	CT								pan-FGFR
Leflunomide	CT								PKC
Melflufen	CT								DNA
Metformin	CT								Complex I
Nelfinavir	CT								Antiviral, Akt
Nirogacestat	CT								γ-secretase
Nivolumab	CT								PD-1
MP0250	CT								VEGF, HGF
ONC201	CT								ERK-1/2
Osalmid	CT								
PD-L1 peptide	CT								PD-1
Pembrolizumab	CT								PD-1
Pralatrexate	CT								RFC-1
Cemiplimab	CT								PD-1
REGN5458	CT								BCMA, CD3
Ricolinostat	CT								HDAC6
Rituximab	CT								CD20
Romidepsin	CT								HDAC
Ruxolitinib	CT								JAK-1/2
Selinexor	CT								Exportin
Siltuximab	CT								IL-6
Sonidegib	CT								Smo
TAK-573	CT								CD38
TJ202	CT								CD38
Tositumomab	CT								CD20
Trametinib	CT								MEK-1/2
Venetoclax	CT								Bcl-2
Vorinostat	CT								HDAC
Aspirin	NA								COX-1/2

The line of therapy of each drug is indicated by numbers according to the frequency of recommendation for clinical use under the National Comprehensive Cancer Network (NCCN) guidelines; drugs under active clinical trial according to clinicaltrials.gov as of 6 May, 2020, are indicated by “CT”; those that did not undergo clinical trial according to clinicaltrials.gov as of 6 May, 2020, are indicated by “NA”. The mechanism of action of each drug is indicated in gray, based on drug labels disclosed by the Food and Drug Administration (FDA) and literature search. The known targets of respective drugs are listed in the rightmost column. For drugs indicated as “CT”, the following criteria were applied in our search: (1) “Condition” was multiple myeloma; (2) “Phase” included phases II, III and IV; (3) “Status” included recruiting, active not recruiting, enrolling by invitation and unknown status. Afterwards, the list of drugs was compiled into a table and those that did not meet any one of the following criteria were excluded from our analysis: (1) active clinical trials, inferred by the last update date lying within 5 years of the date of data retrieval; (2) Phase II and beyond, i.e., for trials in Phase I/II, confirmed progression by literature search to Phase II was required; (3) not included as adjuvant or palliative care in combinations for stem cell transplant, fever and pain management, *etc*.; (4) not included in pan-cancer trials, e.g., MATCH [NCT02465060] (https://clinicaltrials.gov/ct2/show/NCT02465060?id=NCT02465060&draw=2&rank=1), CAPTUR [NCT03297606] (https://clinicaltrials.gov/ct2/show/NCT03297606?id=NCT03297606&draw=2&rank=1) or TAPUR [NCT02693535] (https://clinicaltrials.gov/ct2/show/NCT02693535?id=NCT02693535&draw=2&rank=1); (5) not banned from import in certain countries; and (6) not having multiple records of dose escalation and/or drug combinations in incomplete studies or study termination over the past 5 years, suggesting drug inefficacy.

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
