# Peer review of "Targeting NF-κB Signaling for Multiple Myeloma"

_cancers, 2020, doi:10.3390/cancers12082203_

Round 1

Reviewer 1 Report

The manuscript by Ada Hang-Heng Wong et al. has been considerably reworked and its content has been enriched and reorganized. Overall, the revised manuscript has improved, although some linguistic inaccuracies still remain.

I point out that, for unknown reason, all Greek characters are missing in most of the text.

Author Response

Thank you for your assertion of our hard work. We proofread our manuscript again to correct errors. As we submitted an unformatted manuscript, we notified the Editor for formatting errors.

Reviewer 2 Report

The authors have satisfied all my comments.

Author Response

Thank you for your satisfaction of our response to your comments.

Reviewer 3 Report

This version of the review is significantly improved compared to the previous version. I have only a few minor comments.

line 88: the main target of the IKK-complex is the IkB protein. Processing of p105 to p50 does not üplay a major role. This should be corrected

line 120: likewise here the proteasomal cleavage of the ubiquitinated IkB protein is inhibited

line 138 (and Figure 2):macrophage inflammatory protein is MIP, not MIF (melanocyte inhibitory factor)

Author Response

Thank you for your comments. Here is our response to each of your comments:

line 88: the main target of the IKK-complex is the IkB protein. Processing of p105 to p50 does not üplay a major role. This should be corrected

Thank you for pointing out the inaccuracy. We have corrected the sentence.

line 120: likewise here the proteasomal cleavage of the ubiquitinated IkB protein is inhibited

Thank you for pointing out the inaccuracy. We have corrected the sentence.

line 138 (and Figure 2):macrophage inflammatory protein is MIP, not MIF (melanocyte inhibitory factor)

Thank you for pointing out the typo. We have corrected it in the main text and in Figure 2.

This manuscript is a resubmission of an earlier submission. The following is a list of the peer review reports and author responses from that submission.

Round 1

Reviewer 1 Report

This review is supposed to analyse "Targeting NF-kB Signaling for Multiple Myeloma" but in fact represents a poorly structured compilation of current treatment approaches that, in some instancies, are related to NF-kB signaling. Figures are not really helpful and not well enough described. Although canonical and non-canonical NF-kB signaling are well known nowadays, completely omitting their description is not acceptable (after all this would help to explain the potential role of bortezomib in both pathways). Yet although NF-kB is put central because affected by bortezomib as well as glucocorticoids, these substances have much broader functions, which contribute to therapeutic success.

In my view, the review would need massive rewriting (and probably title change) in order to be acceptable for publication in Cancers.

Reviewer 2 Report

Wong et al presented a manuscript entitled: Targeting NF-kB Signaling for Multiple Myeloma, focusing on anti-multiple myeloma therapeutic drugs targeting the molecular mechanisms of NF-kB signaling.

(1) The authors need to cover in a new section (Section 1), the genetics/genomics and epigenetics of MM, affecting signalling pathways including NF-kB, thereby linking MM to NF-kB signaling.

Having done this, a couple of lines describing these events should be added in the introduction between the first paragraph ending with…..elderly patients [7] and the next paragraph starting with Aside from….

Similarly, this should also be incorporated in the abstract.

(2) In line 7, before …In MM disease progression, the authors should describe briefly the main NF-kB signalling pathways, referring to Fig. 2. The legend to Fig. 2 should also be more detailed.

(3) The legend to Fig. 3 also needs a better explanation

(4) Table 1. Summary of anti-MM drug mechanisms. Again a better explanation is needed. For example, in the first column, there are several lines number 1 in purple, and others numbered 2 or 3 in blue. What is this color coding for?

(5) The legend to Fig. 4 also needs a better description and explanation.

(6) There is a recent excellent review by Eluard B, Thieblemont C & Baud V (2020) NF-kB in the new era of cancer therapy, Trends in Cancer, that the authors need to use for their paper.

(7) In Section 4, the font is not uniform: compare lines 182-224 with lines 225-239

(8) A graphical abstract may also be added

Reviewer 3 Report

In this manuscript, Ada Hang-Heng Wong et al showed that evolution of MM therapies, focusing on anti-MM drugs that target the molecular mechanisms of nuclear factor kappa B (NF-kB) signaling. Also, the authors provide perspectives of contemporary research findings and insight for future drug development. This review is interesting. The manuscript could be further strengthened with a minor revision denoted below.

  1. There are some places that incorrectly or inaccurately write down the manuscript such as page 9, line 225-239 (different font style). Also, authors need to double-check of “references style” because it looks not Cancers format.
  2. It would be better to support the table or graph for information about multiple myeloma patients in the United States.

  3. Add more references in MM therapy part.

    4. It would be better understand if authors add more general information about NF-kB in cancer cells.

Reviewer 4 Report

The paper by Wong H. et al. summarizes the role of the NF-κB pathway in multiple myeloma (MM) and its great impact on current treatment of this disease. This topic is of interest to the scientific community working on MM. Indeed, NF-κB is one of the most important pathways in multiple myeloma not only for its role in pathogenesis, but also for its importance in various treatment strategies. However, authors didn’t extract the logic flow out of the data piles, therefore the general scenario is confusing. To improve the manuscript, it is mandatory to improve its organization and cohesiveness, integrating the relevant literature. Moreover, the manuscript is written in very poor and inaccurate English, which sometimes makes it difficult to understand what the authors want to say. A careful proofreading by a native English speaker is strongly recommended. I also feel that the structure of the review could be much improved.

Here my suggestions and recommendations.

  1. lane 47. Paragraph 1: The title of the first paragraph is not consistent with the text. Indeed, although the title refers to NF-κB signaling, the canonical and non-canonical pathway are not described, while the text is largely dedicated to the interactions between tumor cells and bone marrow microenvironment. These interactions are also depicted in Figure 1.
  2. Figure 2 summarizes the NF-κB signaling; however, activation of NF-κB canonical and non-canonical pathway due to external or internal stimuli is omitted in the text. This is a key point since references to various steps of NF-kB pathways are frequent throughout the text.
  3. Paragraph 2. The authors highlight the difficulty of directly targeting the NF-κB pathway due to its crucial role in immune response, ultimately impinging on infection susceptibility. Then, authors mention different types of therapy that may predispose recipients to infections, jumping from stem cell transplantation to bortezomib and to denosumab, each of which affects immune response against infections through different mechanisms. The logical sequence is missing.
  4. Paragraph 3 deals with MM therapy. I strongly suggest to reorganize this paragraph according to drug category. For example, i. immunomodulatory drugs (IMiDs), ii. Proteasome inhibitors, iii. Monoclonal antibodies, iv. New drugs (such as, IKKB inhibitors, selinexor..), and so on. I also believe that CAR-T directly targeting NF-kB canonical pathway deserve to be discussed.
  5. Table 1 is really difficult to interpret and it is rather inaccurate. For example, lane 1: Bortezomib is a reversible inhibitor of the 26S proteasome and it does not target directly NF-kB. The detailed legend is not useful for interpreting the table and is misleading. It is not clear what does it mean “*True is indicated by green and false is indicated by yellow”. Overall, usefulness of table 1 is really difficult to catch, hence Table 1 needs to be redesigned (or removed at all) and the pertinent legend rewritten.
  6. Figure 4. Neither the text nor the legend allow to understand the meaning of the graph. Moreover, the sentence introducing figure 4 (see lane 142: “Analysis of the recommended drugs along the lines of therapy, together with those undergoing clinical trials, displayed an interesting trend”) does not help to interpret the graph. What kind of trend? What do the percentages shown in the graph indicate? I suggest to remove this figure.

Minor points:

  1. lane 33: “Age demography constitutes significantly to…”, perhaps the authors meant: age demography contributes significantly…..
  2. lane 92: “It is noteworthy that transplant patients….”, should be: It is noteworthy that transplanted patients
  3. lane 195: Recently, Shin, et al, identified lymphocyte cytosolic protein 1 (LCP1) gene as a novel NF-kB target. The word “gene” should be included.
  4. There are several other inaccuracies or inappropriate terms in the paper.
  5. lane 225-339: there is a change in text font style.

Reviewer 5 Report

The Hang-Heng Wong group reports on the effects mediated by NF-KB signaling in MM therapy. The working hypothesis of the group is that NF-KB signaling exerts a control role in the growth of the MM cells in the bone marrow niche where the neoplastic cells grow surrounded by different cell partners.

Fig. 1 and 2 report on the potential interactions among the different molecular pathways. The authors analyze the interactions ruled by the distinct drugs adopted in therapy and the roles likely played following independent sources.

Following this approach, the authors were able to define a sequential story of the effects induced by the different drugs, starting from those used in first-line.

The authors conclude their approach by testing different alternatives completing the original design. Their conclusion is that biomarkers are necessary to monitor the effects induced by the different drugs now in conventional MM therapy.

In my view, the manuscript is well organized and written and the conclusions based on the starting hypothesis. The only limit is that the work may appear sectorial, considering that immunotherapy (and hence the immune response) seems now the most fruitful approach in MM medical treatment.